# Physical and Augmented Dynamics of a Cultural Event

**Naai-Jung Shih \*** and **Tzu-Yu Chen**

Department of Architecture, National Taiwan University of Science and Technology, Taipei 106, Taiwan;
ziyouchen.0617@gmail.com
\* Correspondence: shihnj@mail.ntust.edu.tw; Tel.: +886-02-27376718

**Abstract:** The Taiwan Lantern Festival (TLF) is a specific cultural tradition that has evolved over many years. It is a large-scale festival as determined by the large number of installations and visitors—that is, 20 million visitors in a period of two weeks. The aim of this study is to combine the TLF-related physical dynamics of land use and lantern installations with the augmented dynamics of lantern installations at reallocated sites. We compared five cities in Taiwan with regard to land alterations between 2016 and 2020. The TLF land assessment identified 34 cross-referred types of land use between aerial imagery and GIS surveys in a small area of 2 km × 2 km, in total. The change in land use by year varied between 2% and 499%, up to three times. The complexity of physical dynamics was re-experienced by a more sustainable dynamic of augmented reality (AR) using a scan-to-AR approach to reactivate the installations and fabrics at redeployed sites. The installations of the 2016 TLF were applied. We found that the land use, 3D scan, and AR reshaped the spatio-temporal festivalscape by both types of dynamics. The simulation demonstrated that the fabric retrieved by heterogeneous technologies had equal importance in assessing the host city and in enabling a reactivation for more diversified scales and characters, even with a smartphone AR.

**Keywords:** Lantern Festival; land use; urban; urban fabric; sustainable land management; installations; light detection and ranging (LiDAR); augmented reality (AR); tangible settings; intangible tradition

## 1. Introduction

The Taiwan Lantern Festival (TLF) has expanded beyond a traditional ritual to become an integral part of Taiwan's cultural identity. The TLF is a very popular event that has to fulfill cultural traditions and requires accelerated preparation and construction of tangible settings in host cities, leading to an urbanization process. The "festivalscape" was originally classified as food tourism, whereby festival patrons enjoyed a general atmosphere experience [1], but there is now a new scalable view of TLF that includes settings and tradition. The ever-expanding land-use footprint of the TLF exhibitions has raised questions about the physical form of these festivalscapes and how they are modifying the urban fabric of their host cities.

This study aims to combine the TLF-related physical dynamics of land use and lantern installations with the augmented dynamics of installations at reallocated sites. Five host cities were explored between 2016 and 2020. The festival was assessed according to the urban fabric and as-built festival scenes. The former illustrated modifications made before and after the TLF at each host location following a two-year preparation period. The latter related the TLF scenes to the real urban fabric from a micro point of view as physical feedback that once occurred in the aerial imagery. The TLF study should elaborate and collaborate with heterogeneous subjects and data to enable festivalscape interaction, as a sustainable interpretation of intangible context. In the future, there should be a sustainable redeployment of the festivalscape, which should be conducted collaboratively and interactively between installations, and the land should be reused beyond the TLFs.

### 1.1. Taiwan Lantern Festival

The Lantern Festival has been celebrated during the first full moon of the Chinese Lunar New Year since the Han Dynasty, over two thousand years ago. In addition to lantern exhibitions, the festival includes events that have evolved with different meanings. The government of Taiwan designated the Lantern Festival as a tourism event in 1977 and created the TLF in 1990. Since 2001, it has been one of 12 large-scale folk festivals [2] in Taiwan. It has developed into a festival week, including the three days before and three days after the 15th of January according to the Lunar Year. The urban environment has contributed to different layouts (Figure 1) for the festival in Taipei, where the TLF was originally located. The festival has been hosted by a different city each year since 2003, with each location announced two years prior. It is famous worldwide for its traditions that have evolved according to history, policies, and host-city characteristics over a period of 31 years [3,4]. Although the TLF was canceled or downsized from 2021 to 2022 due to the COVID-19 pandemic, it still attracted 20 million people over a two-week period, with government and non-government agencies recognizing the socioeconomic significance of the festival.

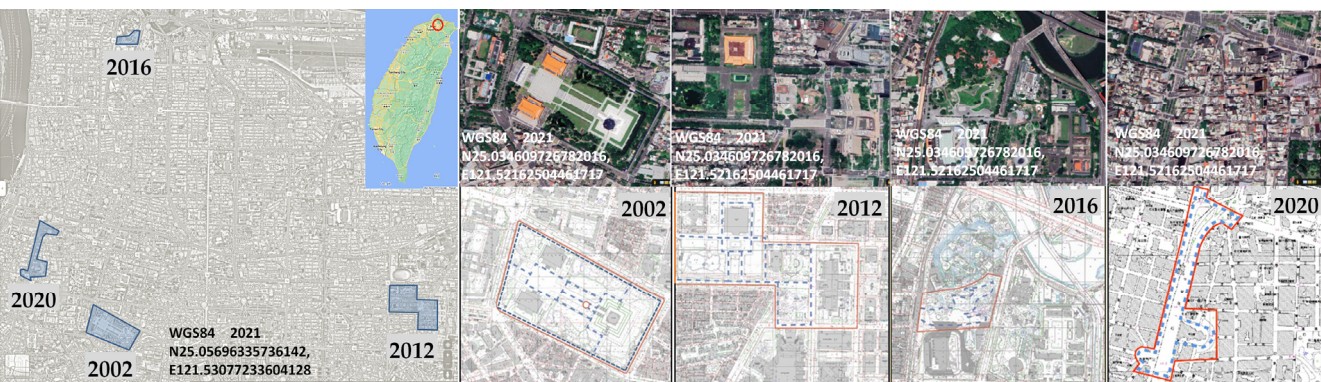

**Figure 1.** Layout of the Taipei Lantern Festivals (2002, 2012, 2016, and 2020), verified from a comparison of satellite images and cadastral maps.

The TLF has evolved over time as a result of interactions between traditions, policies, and the management of artifacts that have been created, displayed, and reactivated. Exhibitions have taken both static and dynamic 3D forms, with displays at ground level, on the surface of ponds or rivers, as mobile parades, and as air shows of unmanned aerial vehicle (UAV) fleets, flying lanterns, and fireworks. Mobile computing has made it possible to extend the TLF experience to the virtual world using augmented reality (AR) and virtual reality (VR) applications both on- and offsite.

### 1.2. Festival and Urban Space

Events are significant creators and manipulators of city rhythms [5]. Incompatible spaces can be juxtaposed with cumulative effects through the transformation of multiple discrete regions [6]. Festivals are considered part of a new heritage paradigm and contribute to urban transformation [7]. Temporal transformation of urban spaces highlights visual culture through the understanding of identity using traditional events and festivals [8].

Events open up opportunities for new interpretations of public spaces [9]. Related spatial organization reveals how a festival can facilitate social interaction at a local scale [10]. For example, art installations when integrated with the urban fabric can be used to rediscover and celebrate the richness of a culture [11]. Lehtovuori compared urban events to a labyrinth [12]. The installations and the urban fabric create a complicated maze of tangible and intangible contexts.

Urbanization and land development are closely connected. Urbanization is a complex social, economic, political, and technological process with no uniform patterns [13].

The urban fabric or current land-cover composition can be changed by transportation infrastructure or military land use [14,15]. The changes can be beneficial or detrimental and can impact landscapes by the different land-use characteristics highlighting the history of the development process overtime.

Urbanization and cultural sustainability are also interconnected. Land use has been considered as part of a sustainable model for activities related to cultural heritage [16]. Land-use changes in an urban area were evaluated for possible impact [17]. Cultural festivals in urban public spaces are exercises in cultural policy [18]. Culture is an important structural factor influencing land systems [19], and the utilization of time and space used by cultural events does raise concerns about environmental impact [20]. The TLF, which represents interaction between culture and the urban environment, involves sustainable urban space, fulfilling cultural sustainability. The "festivalscape" contributes to the physical environment combining event atmosphere and tangible factors [21].

### 1.3. Spatial Data Survey

Zoning changes and the transformation of the urban environment are frequently made for TLFs before and after the two-year period of preparation. The dynamics of local social life can be shaped by special spatial arrangements created for urban festival events [22]. There are numerous similarities and differences that relate to the urban environment, lantern planning, and facilities at various locations. Through the comparison of aerial images, the morphology of the zone distribution and changes in the land use can indicate how the scale can constitute another type of identity that enables the evolvement of the host city. The organization of facilities, installations, and spaces creates different design patterns within open spaces under distribution of distances. Not only can a festival site be connected by commuters from internal and external circulation systems, but the reactivated lanterns can change the urban fabric following the TLF.

Data fusion of multiple heterogeneous sources, with different quality dimensions is part of the lifecycle of remote sensing [23]. Remote sensing systems capture imagery, three-dimensional (3D) geometries, and specific attributes of the physical environment from a distance, either from space, in the air, or on the ground. In the cultural heritage domain, these systems have been widely used to digitally reconstruct historical heritage and cultural landscapes, often at the scale of an artifact, a building, or an archaeological site, for preservation and quantification. Remote sensing and geographic information system (GIS) data are usually integrated [24]. Light detection and ranging (LiDAR) studies have been conducted for city reconstruction [25], physical–cultural heritage [26], and AR applications [27–29] in real environments. Remote sensing imagery can be integrated with LiDAR data to improve geopositioning accuracy in metropolitan areas [30]. The integration of aerial imagery, 3D laser scans, and GIS has been applied in different domains under various emphases [31,32]. The integration has improved the efficiency of data representation and enriched cultural events even when applied to temporary installations.

## 2. Materials and Methods

Spatio-temporal dynamics affect the urban fabric and installations before and after holding a festival. The intensity of construction and the number of visitors strongly impact the urban environment during a limited period of time. Whether an event is derived from the short-term evolution of land use can be investigated with data, such as aerial imagery and LiDAR data. Further investigations should be conducted at the city scale by examining the impact of cultural events on the urban landscape and by using aerial imagery and LiDAR to examine the urban fabric of TLFs, from a macro-perspective and the as-built installations from a micro-perspective.

The festivalscape, installations, and urban fabric contribute to the formation of an interconnected description of an event. When a festival occurs on an island-wide scale, the urban fabric can be modified temporarily or permanently, and this can be confirmed by comparing the situation before, during, and after the event. This study used aerial imagery,

3D scanning, and AR to capture and recreate the festivalscape. Satellite images and GIS maps, which were used for urban planning and development, provided a timeline of urban fabric evolvement before and after the changes (between 2016 and 2020). Three-dimensional scans were used to document the fieldwork conducted for the 2016 TLF in Taoyuan, which provided a more detailed as-built description. Both types of data had scalable levels of detail that were used for spatio-temporal comparisons of the festivalscape.

### 2.1. TLF from Physical and Augmented Dynamics

The space layout of the Lantern Festival usually consists of an interlaced circulation system of installations, display stages, and visitors. The zoned display is similar to a museum, but with no walls. Cross-referencing between tangible measures (for example, land-use changes and lantern installations) and the intangible festival was performed using satellite images and as-built constructions. Historical imagery data were examined to determine the extent of the modified surface area and the speed at which it was modified.

Verification of the urban fabric dynamics consists of physical and segmented entities, the 2016 TLF as-built settings, and interactions with the urban fabric (Figure 2). The reconstruction part of the acceptance check was partially feasible for physical items from a micro-perspective, but from a macro-perspective there was limited data availability in the wider area for the as-built planning setting. In addition to the physical objects and site layout, the festive ambiance of the physical environment, as described by the festivalscape in this study, was determined from the reconstructed 3D scenes from the evening.

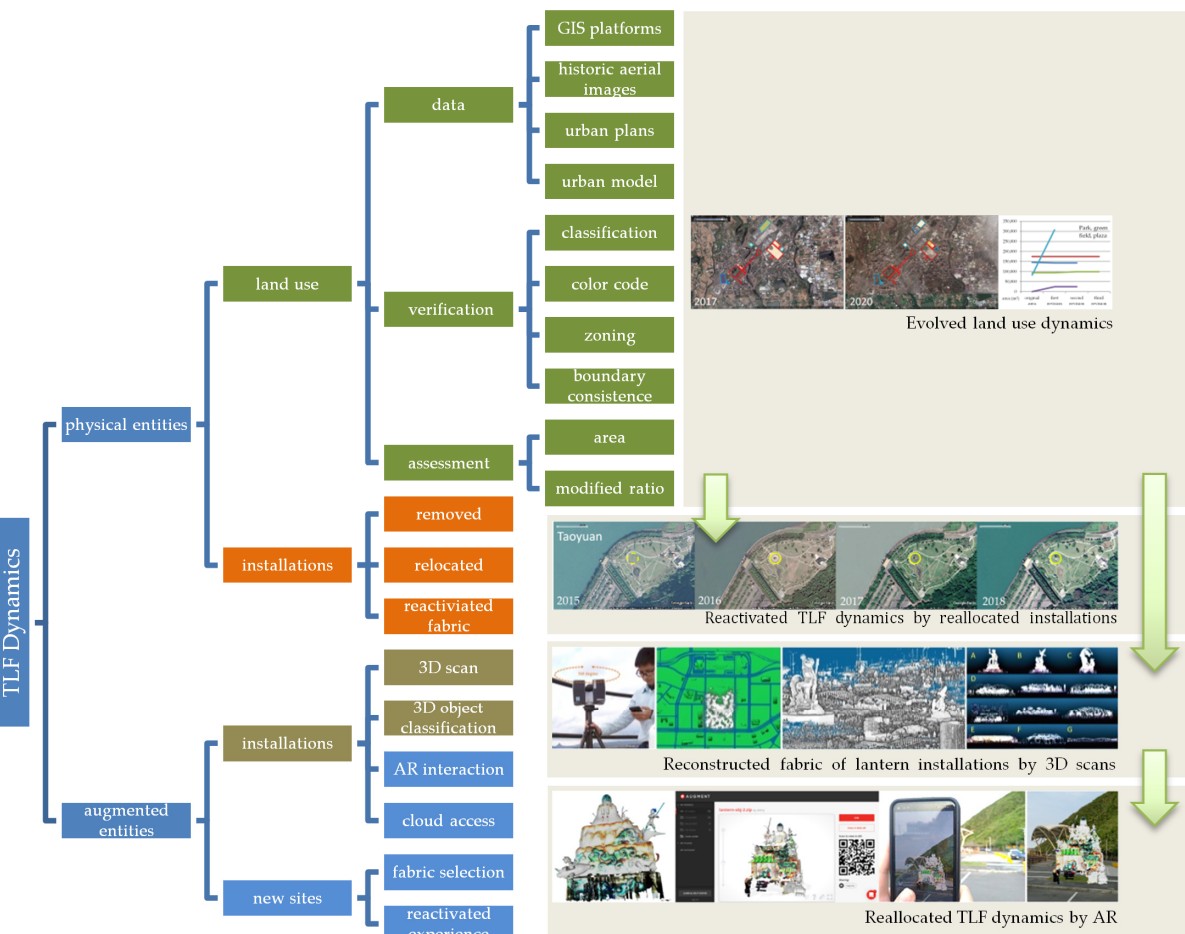

**Figure 2.** Research framework and flowchart.

Reverse engineering is generally applied by checking the tolerance between the as-built product and original design to ensure production quality [33] or through the assessment of

the urban fabric for the purpose of modification confirmation, analysis, management, or illustration [34–36]. The representation of a festival can be performed in the same manner to document the tangible part of the lantern-related installations in 3D or the physical appearance of the festivalscape using crowds, behaviors, and interactions as references.

In order to associate the satellite images with the as-built description of a TLF, an AR platform was applied to allow 3D installation models to be simulated on different backgrounds as an extension of experiences from remote sites. Follow-up developments on festival reinstallations were also performed.

### 2.2. Geospatial Data Sources and Referencing

Five types of maps, images, and plans were applied (Table 1): (1) Taiwan Geospatial One Stop (TGOS), Urban and Rural Development Branch, Construction and Planning Agency, and Ministry of the Interior (MOI) [37]; (2) Taiwan Map Service, National Land Surveying and Mapping Center (NLSC), and MOI [38]; (3) Google Earth Pro®; (4) Google Maps®; and (5) urban plans [39–41]. Maps were already georeferenced in the GIS platforms of TGOS and NLSC.

**Table 1.** The five types of maps, images, and plans used.

| Source | Data Type | Years | Coordinate Reference System (CRS) | Resolution |
|---|---|---|---|---|
| TGOS | Official land-use maps | 2015, 2018, and 2020 | TWD97(EPSG:3826) | 0.298 m in 1/1000 |
| NLSC | District land-use maps | Between 2005 and 2020 | TWD67, TWD97, TWD97[2010] | 0.298 m in 1/1066 |
| Google Earth Pro® | Satellite and aerial imagery, GIS data | Between 2015 and 2020 | WGS84(EPSG:4326) | 0.1~0.2 m |
| Google Maps® | Satellite imagery, topological maps, vector graphics | 2020, 2021 | WGS84(EPSG:3857) | 0.149 m in 1/533 |
| Taoyuan | Urban plans | 2021 | TWD97(EPSG:3826) | |
| Yunlin | Urban plans | 2021 | TWD97(EPSG:3826) | |
| Chiayi | Urban plans | 2021 | TWD97(EPSG:3826) | |

Satellite or aerial images were retrieved and analyzed to examine the evolved urban fabric according to newly added or modified land uses. Image capture of the development was performed prior to, during, and after the festival. Although the exact images taken during the TLF may not have been found, the constructions, which may have started two years before the event, revealed the interactions between the festival and the surrounding urban fabric.

TGOS information from 2015, 2018, and 2020 was referenced to check the zoning modifications made to the land-use classification and to determine whether matches were made between construction types and classifications. Each type of land use was assigned by a specific color on a map. The color-coded area was calculated to quantify the development of the festival site. District maps of land use were retrieved from the NLSC, and maps between 2005 and 2020 were compared. Due to an inconsistent color code system for land use between 2006 and 2019, cross-referencing was performed with the TGOS to obtain updated data.

We consulted historic images from Google Earth Pro® to identify inconsistent areas based on the color code between 2015 and 2020. Images taken between 2002 and 2009 were missing. The most current aerial Google Maps® images were compared with historic Google Earth Pro® images and used to identify changes made between 2020 and 2021. The first and second layers of information in the urban plans were applied to specify the evolved revisions in the defined area-of-interest cited in the urban plans initialized by the related local government planning departments.

Satellite images, vector drawings, urban plans, specifications, and 3D-point clouds presented different levels of tolerance in defining the boundaries. The images were applied from different Google® platforms, which were prepared and integrated with maps surveyed in different years with periodic updates. We registered urban plan drawings with satellite images to derive exact land-use boundaries, allowing us to determine land-use changes. Additional verification was conducted, since the color code was not consistent by GIS platform or year at the same location. The level of precision measured by area and distance was not confirmed from the provider and was used for reference only, since the maps came in different coordinate systems and went through a series of projections, overlays, and transformations between the systems.

The estimation of the size of the individual land-use area consisted of several steps. The land size was first verified by the Google® measurement tool. Since differences existed between the map of the land-use district and the satellite or aerial imagery, the imagery was referred to for the correct estimation of area, and then the boundary filled with the corresponding color code defined by TGOS or NLSC. After the boundaries were defined, the area of the same polygon set was automatically estimated by selecting its color code. A parking area was temporarily assigned to a space nearby the TLF. Its area, which could not be found by the color code of land use in the imagery, was estimated by a festival map, news report, and blogger webpages.

### 2.3. 3D Scan of Installations and Facilities

LiDAR was used to capture 3D as-built scenes with accurate dimensions and relative locations on the 2016 TLF in Taoyuan for verified matches between segmented point clouds and the categories that a TLF was made of, for example, the main lantern and auxiliary lantern design. The historical 3D scanned scenes, which were produced using a Faro® laser scanner (Figure 3), were re-applied to integrate and display as-built scans and the temporary or permanent urban fabric of crowds, installations, and buildings. The scanner applied a time-of-flight (TOF) algorithm to calculate its distance from the target and created a 3D configuration of the surface. It conducted scans of 360° horizontal and 305° vertical fields of view along a planned path. In total, 62 color scans were performed in ASC format and registered automatically by a contractor, Hsien Fa Enterprise Co. Ltd., Taoyuan, Taiwan, using Faro Scene 5.0®. The scans were made in the evening. The size of the files added up to 24.02 GB. After receiving the scan data, we subsampled, segmented, rendered, or reformatted the point cloud as a standard procedure [42]. All data were separated into several smaller sized files. The point cloud was also converted to a mesh model of AR interactions using the Augment® platform.

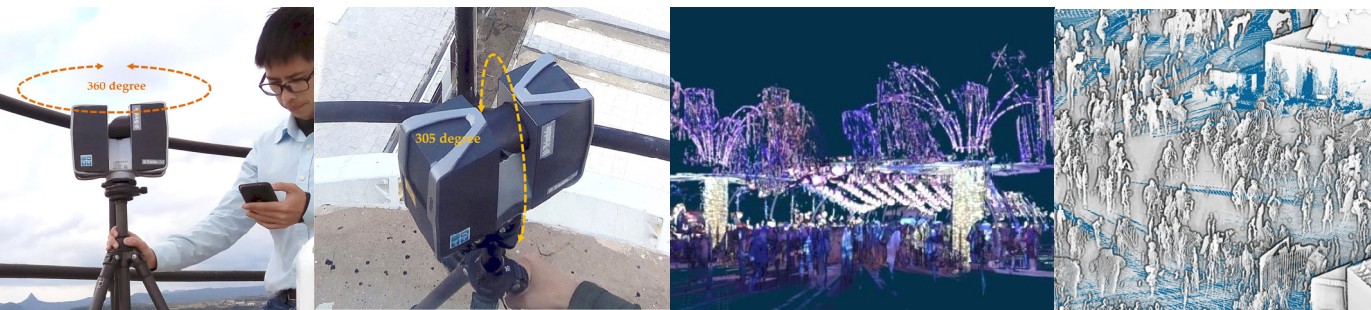

**Figure 3.** Faro Focus 3D® laser scanner and field scan.

As-built 3D scans created detailed as-built descriptions and self-explainable configurations of the temporary urban fabric. Related visualization methods created scalable detail that integrated the urban fabric and the 3D lantern installations. The full-scaled data also created a reference scale to connect temporary modifications to the permanent setting. Post-3D scan data processing included tender projects quantification, AR simulation, and installation reactivation.

## 3. Results

In total, this study involved five cities (Taoyuan, Yunlin, Chiayi, Pingtung, and Taichung), five TLFs (2016–2020), two official GIS systems (TGOS, NLSC), four types of information (satellite imagery, cadastral maps, annual surveying and mapping of land-use districts, urban planning reports), six years of geographic information (2015–2020), 50 satellite imagery, five cities' urban planning databases, five land-use district maps, and 151 land-use color codes. An area of about 70,000 $m^2$ was 3D scanned.

The differences between remote sensing and 3D scanning were examined to show the spatio-temporal evolvement of the urban fabric. Over a period of five years, 34 kinds of land-use changes in five cities occurred within a total area of 4,479,910 $m^2$, roughly 2116.6 m by 2116.6 m. The frequency of change was directly and indirectly related to the planning of the TLF. Modifications of the area were performed to accommodate complicated events, and quantitative descriptions were recorded of areas as small as 30 $m^2$ ("religion"). From the size of a city down to the size of an installation, a quantitative description of the land use was applied.

### 3.1. Aerial Image and GIS Verification of Transformed Urban Fabric

The refabrication and recovery of urban areas represented the interventions applied to the local environments (Figure 4). Based on GIS surveys, vacant land and unused land were frequently used for the TLFs. Vacant land includes land under preparation or being developed for a specific use. Unused land, which is not designated for a specific use, is one of the subcategories of vacant land. Constructions that were newly added to or recovered from old settings or circulation systems are illustrated in the aerial images.

### 3.1.1. Yearly Revision of TLF Sites

Segregation and combinations were made repetitively to various land uses before and after the TLF. A review of each TLF site was conducted comparing unaltered and new/altered areas. The estimation of the 34 types of land uses was used to form a detailed quantitative description with newly appeared or disappeared land-use types, for the areas shown in Figure 4. Although the government initiates a survey every year, the island-wide area is too large to be completed in one year. Therefore, survey data are not available for every year at each TLF site.

Each change was checked and calculated. The patterns were analyzed and characterized as follows.

- Unaltered: Land use remained the same before and after the TLF, for example, "Park, green field, plaza", "Street", and "Religion" in Taoyuan.
- Altered: Land use changed, for example, from 2015:"Under construction" to 2016:"Cultural facilities" in Chiayi.
- Segregation or combination: Land use was divided or involved the combination of more than two types. For example,
  - ○　2015:"Unused land" = 2016:"Dry farmland" + 2016:"Vacant land" in Chiayi;
  - ○　2015:"Gas station" + 2015:"Retail and wholesale" = 2016:"Public facilities" in Chiayi;
  - ○　2017:"Government" was divided into 2019:"Park, green field, plaza" and 2019:"Streets and facilities" in Taichung.
- New: Reclaimed land, for example, 2017:"Grassland" in Pingtung.

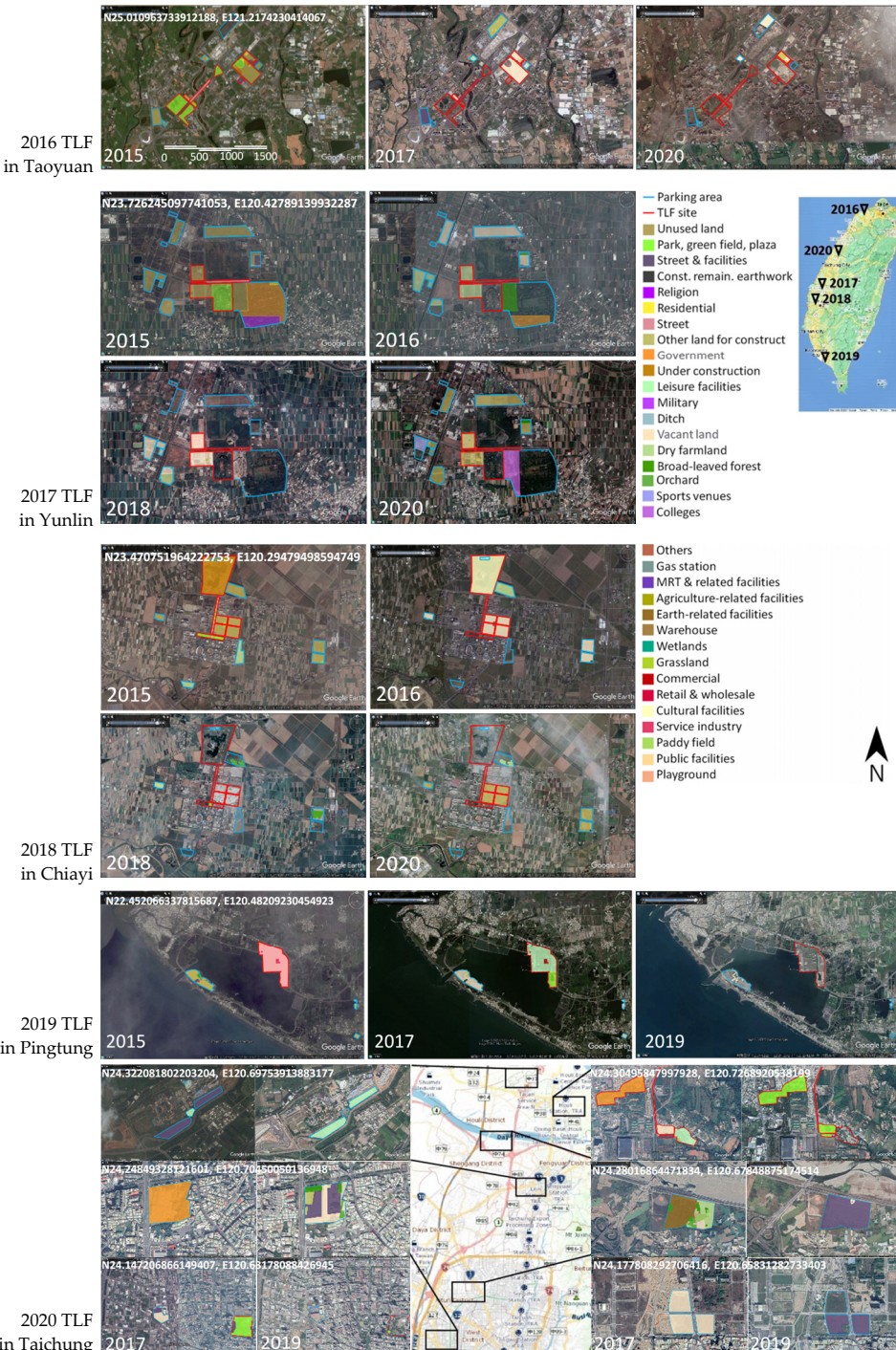

**Figure 4.** Satellite images illustrating the evolved urban fabric before and after the 2016 TLF in Taoyuan, the 2017 TLF in Yunlin, the 2018 TLF in Chiayi, the 2019 TLF in Pingtung, and the 2020 TLF in Taichung (from the top to bottom).

The complexity can be explained by the areas of specific land use (Figure 5) and the total percentage changes (Figure 6). The areas of specific land use showed a stable common land-use status, which was also shared by many local events as altered uses. The total percentage changes provided a summary of the relative scale of development. The three types of land use shown in Figure 5 maintained similar configurations of land use over time, except for the 2020 TLF in Taichung, where steep increases in the areas of "Park, green field, plaza" and "Street and facilities" were observed, even with short-term preparations.

This was a unique instance, demonstrating how the TLF was an instrument that modified the urban fabric, among other types of land-use change.

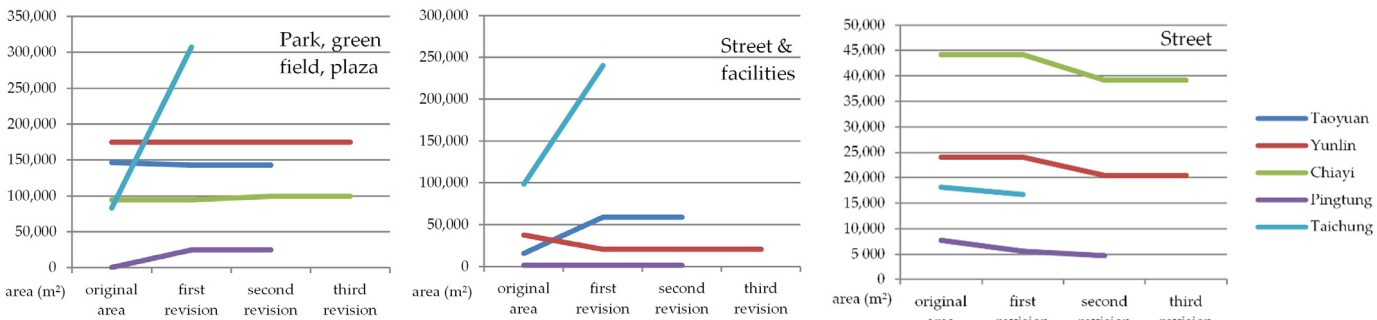

**Figure 5.** Yearly revision of the TLF site by specific land uses.

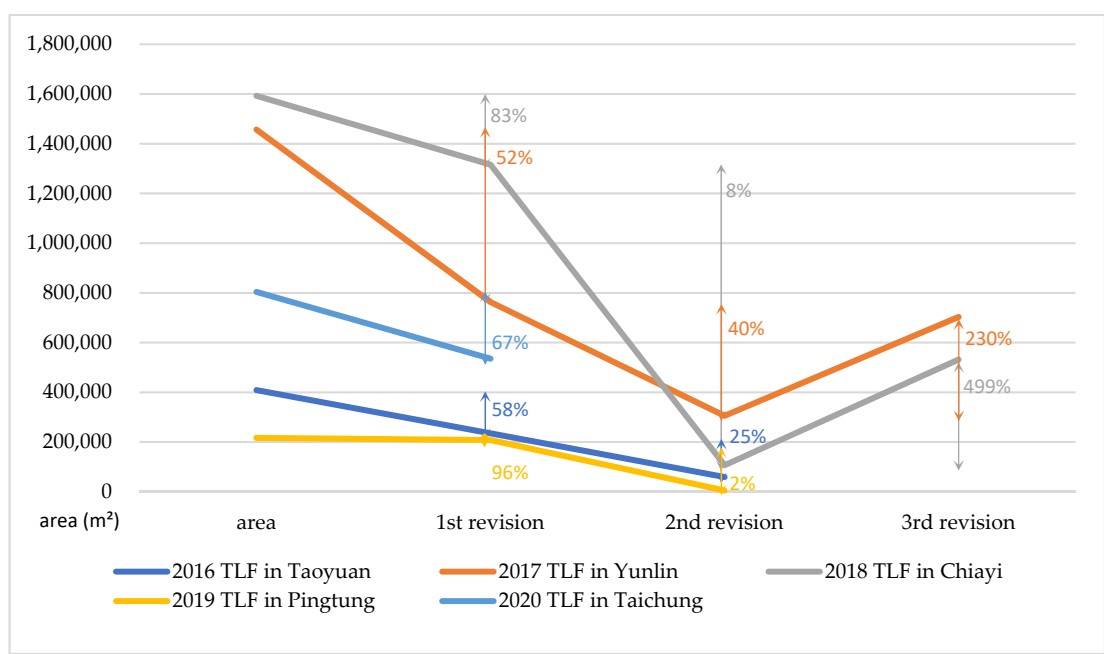

| | area | first revision | second revision | third revision |
|---|---|---|---|---|
| 2016 TLF in Taoyuan | 2015 | 2017 | 2020 | |
| | 408,625 m² | 235,065 m² | 59,512 m² | |
| | | 58% | 25% | |
| 2017 TLF in Yunlin | 2015 | 2016 | 2018 | 2020 |
| | 1,456,917 m² | 762,765 m² | 305,476 m² | 703,346 m² |
| | | 52% | 40% | 230% |
| 2018 TLF in Chiayi | 2015 | 2016 | 2018 | 2020 |
| | 1,592,536 m² | 1,315,430 m² | 106,626 m² | 531,645 m² |
| | | 83% | 8% | 499% |
| 2019 TLF in Pingtung | 2015 | 2017 | 2019 | |
| | 216,848 m² | 207,970 m² | 4988 m² | |
| | | 96% | 2% | |
| 2020 TLF in Taichung | 2017 | 2019 | | |
| | 803,782 m² | 535,200 m² | | |
| | | 67% | | |

**Figure 6.** Yearly revision of TLF sites by total area and percentage.

A revision of each TLF site by year was conducted, with the percentage of area change being between 2% and 499% (Figure 6). Although the absolute area varied, the percentage increased after the 2017 TLF in Yunlin and the 2018 TLF in Chiayi in the third revision. Most of the land changes were from unused to vacant land, and both types were usually

undeveloped. The urban fabric remained similar in 2016, 2017, and 2018, as revealed in the satellite imagery. Urban planning in the land-use districts for the 2016 and 2018 TLFs was defined in the "Special District Plan of the THSR Taoyuan, Yunlin Station" and the "Urban Planning of Chiayi County". The missing line segments shown in Figure 6 indicate areas where no revisions were made.

### 3.1.2. Impact of Over-Mobility on the Provisional Fabric

The cityscape was included by the government as one of the nine tourist service blueprints for developing a large-scale event [43], with the key to the success of an event being determined by the transport infrastructure [44]. Events can have significant impacts on communities and their economies [45,46], activities [47], and recreational and educational opportunities [48]. Over-mobility, as the origin of "over-tourism" [49,50], occurs when the number of tourists is greater than the number of local residents or carrying capacities of the festival and surround area. This has negative consequences, such as an increase in traffic, loss of identity, and a decreased quality of life [51]. The accessibility and mobility of cultural services and social facilities should be improved through more efficient transportation and less pollution [52] and by providing maximum satisfaction [53].

One of the major differences in land use came from the planning of parking spaces, illustrating the impact of mobility on the host city due to local and island-wide tourists. In 2019 TLF, temporary parking lots were added next to the site where the international leisure district was planned without sufficient parking spaces. The 2016 TLF site, however, was located just in front of the THSR station and had much easier island-wide access.

Three types of land transformation were observed:

- Recovered to original fabric: temporary fabric was removed, except for ground pavement.
- Remodeled fabric: for example, a new building was added to a former festival site in Yunlin County.
- Replotted to a new zoning type: this was similar to the first condition, with changes made only to zoning. Verifications were made using cadastral maps.

In the investigation of physical dynamics, the scalable representation of satellite images and 3D scans facilitated the interpretation of over-tourism, as the scattered parking lots (Figure 4) and throngs of visitors can be seen from 3D models of the streets and in the lantern sites (Figure 3). We found that the construction of temporary parking lots and the selection of sites next to a THSR station represented two different instruments that could promote sustainable consumption during the TLF.

### 3.1.3. Fabric Reactivation by Lantern Exchange and Relocation

Sustainability, measured according to social, economic, and environmental factors, was exemplified by fabric reactivation and lantern reinstallation. Physical dynamics enabled the exchange of installations domestically and internationally. Cultural exchanges were conducted overseas in 2016 with exhibitions of two large lanterns and six small traditional ones in Hong Kong and Kagawaken, Japan, respectively. After the 2020 TLF, 21 lanterns were collected by local organizations. Although construction rapidly changed the urban fabric, reinstalled lantern designs also modified the cultural and urban fabric in different locations. Some of the relocated installations can be visually identified from satellite images (Figure 7). As a result, the reinstalled lanterns not only temporarily changed the urban fabric during or after the TLF at the same site, but also reactivated the fabric at different locations permanently.

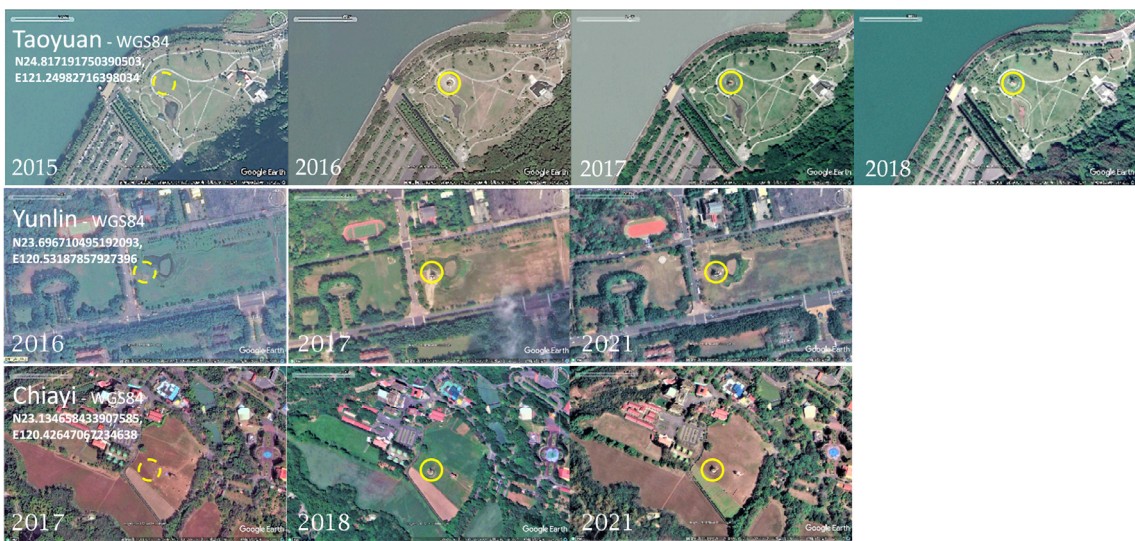

**Figure 7.** Relocated and reinstalled themed lanterns in satellite images.

### 3.2. 3D Scans of the 2016 TLF in Taoyuan

Three-dimensional scans of the 2016 TLF in Taoyuan were used to verify the placement of lanterns and facilities in the festivalscape. Due to time and budget constraints, only the main region (Figure 8) and the left part of the Blue Pond were scanned. After a detailed review of the model and rendered images, items other than self-explainable configurations were visually identified, for example, small lanterns, workers, banners (on street light poles), pavements (by referring to the satellite images before and after), and traffic guides/fences/controls/signage/barriers. All of the detailed items were distributed among various tender projects with partially identifiable physical entities presented visually.

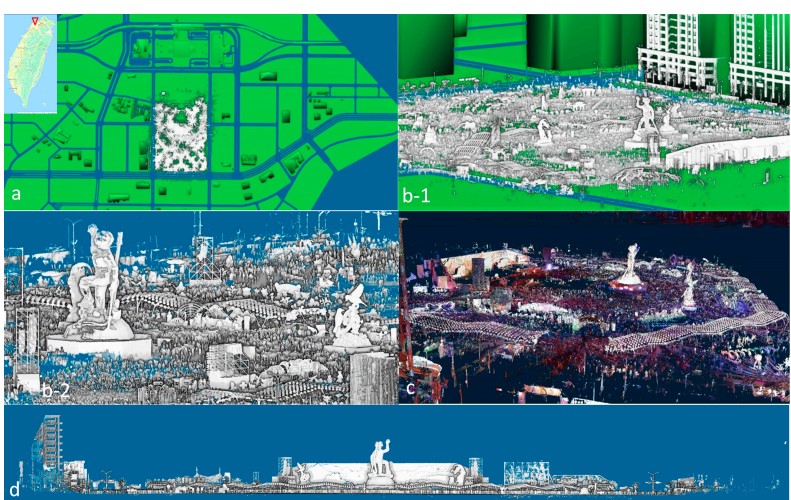

**Figure 8.** (**a**) Reconstructed festival scene and existing urban model, (**b-1**) perspectives of the point-cloud model in plain mode, (**b-2**) a closer view of the model details, (**c**) perspective of the point cloud in original color, and (**d**) elevation with the presentation of crowds at ground level.

The identification of installations was achieved by the 3D-point cloud to cross-reference configurations of segmented 3D models (Figure 9), a description of installations (Figure 9b,c), and the festival map. The proportion of other installations and those closely related to the main site and the Blue Pond site was approximately 2:1 (67% to 33%) (Figure 10). For 33% of the items, the 3D physical form enabled the representation of settings measured in 75% of the items from the 3D-point-cloud model. The unscanned parts, 25% of unidentified items, comprised projects held in the preparation stages, which were broken down into tasks or

objects that were too small to identify, operated behind the stage, or were blocked by other objects; items applied in other lantern areas; the dynamic projection of media; maintenance; or activities held at times other than when the scans were performed.

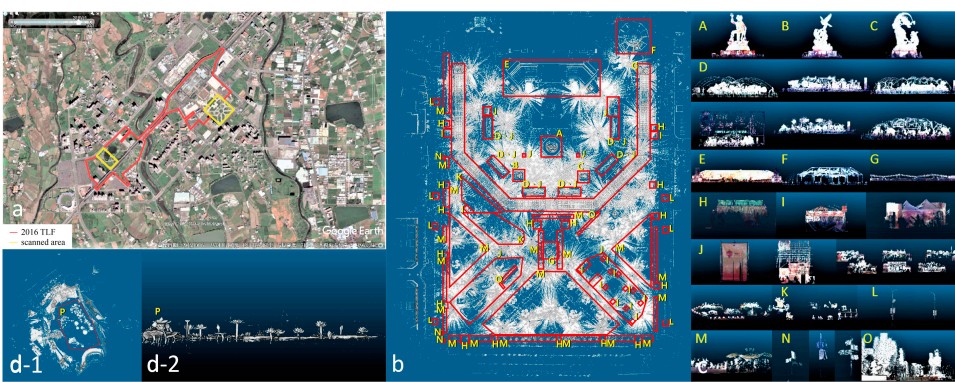

**Figure 9.** (**a**) Scanned boundaries, (**b**) 3D-point-cloud model in the main area with lanterns and facilities labeled alphabetically, (**c**) segmented parts, (**d-1**) the Blue Pond and the set of lotus lanterns on plan view, and (**d-2**) lanterns on elevation view.

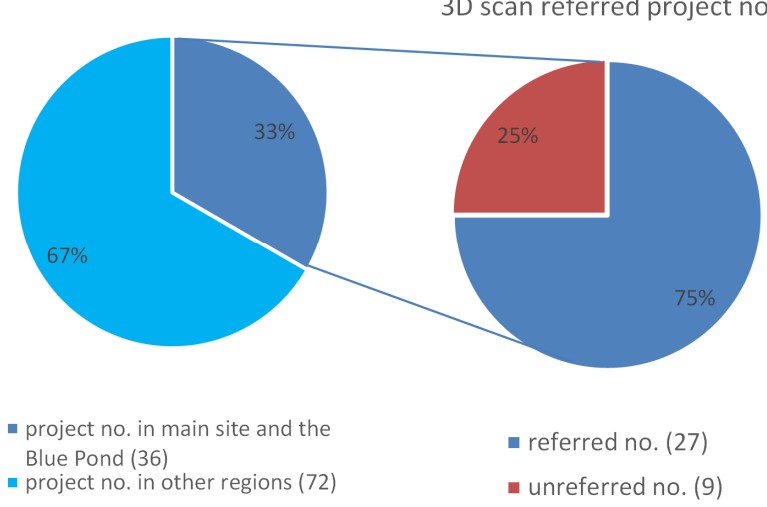

**Figure 10.** Distribution of project numbers between the main region, the left part of the Blue Pond, and the other areas (**left**); and tender projects identified from the 3D-point-cloud model (**right**).

### 3.3. AR Dynamics

AR interactions were applied to communicate and relate the findings made from satellite images and field scans by re-experiencing the installations at redeployed sites or scenes. Figure 11 provides an example of the application of two AR models from the Augment® platform to eight locations. The point-cloud geometries were retrieved in the as-built form. In addition to the recycling of physical lantern creations for next year, another type of dynamic could be achieved from the scenes presented in digital format. AR mesh models can be viewed through an internet platform, such as Augment®. The associated model and interactions, which represented the liveliness of old festivals, provide the possibility of reactivating former experiences within the existing urban fabric. Although field experience has been limited recently due to the COVID-19 pandemic, reinstallations of reconstructed objects have been made by scanning the AR code to simulate the scenario on the waterfront.

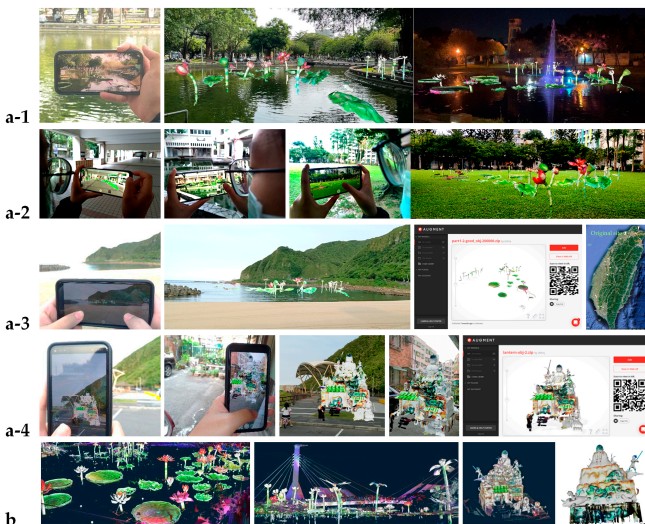

**Figure 11.** AR field experiences were simulated by applying two AR models from the 2016 TLF on the Augment® platform at new sites, during the day and night: (**a-1**) in a park, (**a-2**) on a campus lawn, (**a-3**) on a beach, and (**a-4**) on a street; (**b**) original color point-cloud model.

## 4. Discussion

Events can be regarded as an engine for regional growth in terms of tourism, services, and infrastructure [54]. The transition of land use illustrates the complexity of urbanization [13] that has occurred over 70 to 180 years [13–15,55] under the dynamics of a matrix [56]. The preservation of cultural festivals has involved definitions that were frequently assigned to planning settings, installations, and layouts. In this study, the festivalscape was visually, qualitatively, and quantitatively described using an approach based on the heterogeneous combination of as-built data and documentation for physical and augmented dynamics. The short-term development pace created interactions between large-scale events and the urban fabric, as well as fulfilling the preservation of settings and tradition. We found that urban evolvement can be quickly accelerated by the special event after approximately two years. Based on short-term development, land use presented location-dependent development patterns. Local fabric can benefit from construction and the recovery from festival-related remodeling works in three dynamic types of land transformation.

Evolvement can be made from the TLF site, measuring approximately 2 km × 2 km, to the entire island only by using augmented dynamics. Land-use assessment was usually hidden or presented in a limited way to the public. This study demonstrated that the physical and temporal dynamics (land use, fabric) are meaningful and can be reactivated in augmented dynamics (installations) in three ways: area assessment to evolving trend, from land use to installation, and from installations to remote reactivation, in three tracks, according to the arrows presented in Figure 2.

The remote sensing data, 3D scan, and AR were interconnected on a cultural basis in this study, from local landscape, temporal installations, to situated interaction. Dynamics were initially reactivated by physical dynamics from land-use alterations, reallocation of installations, to reactivated fabric. Reactivated physical dynamics, which was land-use dependent, was connected to installations with micro-scaled experiences. Reactivated augmented dynamics enabled the virtual reallocation of the installations. The reactivated fabric and formal TLF models presented micro-scaled experiences which were independent of former land-use types.

The type of fabric modification revealed the dynamics in the pace of a cultural event. The 34 land-use types shown in Figure 4 represent the complexity of the urban fabric between 2015 and 2020. The revision of the TLF site by year revealed changes that varied between 2% and 499%; some areas underwent as many as three revisions. For example,

2017 TLF in Yunlin had part of the vacant land transformed into a stadium, as shown in the 2020 satellite image by two sector-shaped pavements. The 2018 TLF in Chiayi had part of the "under construction" land in 2015 transformed to "cultural facilities" in 2016. The building and pavement were under construction in 2015, and the space layout and facility were almost complete in 2016.

TLF incorporated different scales and characters of dynamics from land use to installation, from physical to digital, and from physical to augmented. Moving from physical to augmented dynamics enabled the connection from land-use dynamics to personal experience, as an updated manner of reactivation. Large-scaled dynamics in the host city is converted to an evolved augmented scale and dynamics in any reallocated manner. The reallocation of installations, which was considered as a sustainable act of the TLF, is replaced by augmented dynamics and becomes more than a fixed entity presented in aerial images.

Scales and dynamics were interconnected, in which festivals were scaled up to multiple land uses, scaled down from remote sensing to 3D scan, and further transferred to digital form suitable for situated applications. This technology advantage, which is available to scale down from festival to personal experience, facilitates the dynamics between cloud-accessed installations and new fabrics of various scales. The interaction presented a new interpretation of the cultural event and further promoted it in an evolved scale and manner.

Table 2 presents the differentiated characteristics between the physical and augmented dynamics. The evolving trend includes the practice of a new augmented identity in an enriched cultural landscape, or a new enriched cultural identity in an augmented landscape anywhere. The physical-to-augmented dynamics were inspired by a city-wide site distribution. The simulation can be applied in a wider area by mobile AR. The augmented-to-physical dynamics were expected to enrich the content yearly and to assist the cultural preservation for future activation. More diversified allocation of AR should enable local-identity enhanced-augmented tourism.

**Table 2.** Differentiated characteristics between physical and augmented dynamics.

|  | **Physical Dynamics** | **Augmented Dynamics** |
| --- | --- | --- |
| Land use | Host-city dependent | Situation dependent |
| Installations | Temporary | Cloud access |
| Order | Land use goes before installation | Equally involved |
| Festivalscape | From macro- to micro-representation | From micro- to macro-representation |
| Involved database and devices | GIS@5, 3D scanner | Augment® AR platform and cloud database, smartphone |
| Measures | Areas, land-use types, no. of revisions, distribution (over mobility, broad allocation of lantern zones) | No. of AR models, size of database, applied scenes and locations, geographic distribution of applications |
| Occurrence | By cities annually | Free of rotation |
| Scales of dynamics | 2 km × 2 km | Island-wide or worldwide |
| Involvers | Group collaboration and exchange | Individuals |
| Timing of execution | Two years ahead of TLF + after | Unrestricted |
| Duration of preparation | Two years ahead of TLF | Days for 3D scan and model conversion |
| Development Pace | Short-term, by rotations | Long-term |
| Geographic distribution | Host-city centered | Island wide |

**Table 2.** *Cont.*

|  | Physical Dynamics | Augmented Dynamics |
|---|---|---|
| Visitor perception | More installations than land use | Both |
| Analysis | Quantitative description | Qualitative description |
| Concurrency | Festival-dependent, all festivals for two weeks island-wide | Festival-independent, on demand anytime |
| Hierarchy | TLF at top hierarchy | Hierarchy independent |
| Identity | Local identity | Cultural identity |
| Budget | High: land, installation, reallocation | Low: cloud-accessed database, public cloud platform |
| Reactivation | Integrated cultural landscape, distributed identity, diversified reactivation | Smartphone enabled reactivation, cloud access |
| Cultural reactivation | Reallocation and exchange of installations | Situated AR application |
| Future promotion | By exchange occasionally | Long-term application |

Table 3 presents the evolving loop in festival adaption according to three tracks. The action for Track C facilitates the simulation of participation and atmosphere from re-experienced fabric, to reduce mobility, reinstall donated lanterns, entertain visitors, incorporate design, and promote identity.

**Table 3.** Evolving loop in festival adaption.

| Subjects | Timeline/Typology of Activation | | | Actions |
|---|---|---|---|---|
|  | Track A | Track B | Track C |  |
| Installations | Old | Old | AR | Recycled, reinstalled |
| Fabrics | Old | From old to new | New | Replotted |
| Presence of environment | Physical | Physical | Augmented + physical |  |
| Interaction | Walk-through | Walk-through | Augmented walk-through |  |
|  | Host-centered landscape | Reallocated landscape | Improvised/situated landscape |  |
| Level of changes | Maximum | Medium | Minimum |  |
| Involved maximum | Maximum | Medium | Minimum |  |
| Cultural contents | Initiation | Reactivation | Reactivation by interaction |  |

## 5. Conclusions

Cultural assets can take tangible and intangible forms that operate bi-directionally, co-dependently, and non-linearly [7]. As outlined in Taiwan's 1982 Cultural Heritage Preservation Act [57], a clear understanding of these issues could help TLF to evolve as a means of sustaining culture. We found that the festival-related fabric in each city presented significant differences either from the original urban plans or from zoning replots applied afterwards. Urban evolvement can be quickly accelerated by a special event after approximately two years. Based on short-term development, land use presented location-dependent development patterns. We could quantify the measures of the enlarged event scale in sites that measured approximately 2 km × 2 km.

The looped evolution of fabric extended remote sensing systems to a city scale to examine the land-use impacts of the TLFs on the urban fabric from the macro-perspective

and the as-built installations from the micro-perspective. We found the AR platform provided a novel contribution to extend or reshape the former festivalscape in a related, yet not exactly the same, manner. The festivalscape experience was distributed beyond the original geographical boundary to new topological locations.

The complexity of physical dynamics was re-experienced by a more sustainable dynamic of AR using a scan-to-AR approach to reactivate the installations and fabrics at redeployed sites. AR in the study of TLF presented a converged, re-experienced, and redeployed context of culture and fabric. We concluded that this is a feasible strategy for representing interactions. The spatial structure combines the advantages of heterogeneous data for reactivated experiences of culture. Future research would explore diversified cultural-related dynamics to enrich the interaction with the urban fabric.

**Author Contributions:** Conceptualization, N.-J.S.; methodology, N.-J.S.; 3D-scan software and AR models, N.-J.S.; AR software field interaction, T.-Y.C.; validation, N.-J.S. and T.-Y.C.; formal analysis, N.-J.S. and T.-Y.C.; investigation, N.-J.S. and T.-Y.C.; resources, N.-J.S.; data curation, N.-J.S. and T.-Y.C.; writing—original draft preparation, N.-J.S. and T.-Y.C.; writing—review and editing, N.-J.S.; visualization, N.-J.S. and T.-Y.C.; supervision, N.-J.S.; project administration, N.-J.S. All authors have read and agreed to the published version of the manuscript.

**Funding:** This research received no external funding.

**Institutional Review Board Statement:** Not applicable.

**Informed Consent Statement:** Not applicable.

**Acknowledgments:** The information applied in this study can be accessed from Taiwan Geospatial One Stop (TGOS), National Land Surveying and Mapping Center (NLSC), Google Earth Pro®, Google Map®, Maxar Technologies®, and urban plans of each city and county shown in the References section. The trademarks of Google Earth Pro® and Google Map® are displayed at the bottom of each map to illustrate the origin of the source. The authors would like to express a sincere appreciation for their supports and all the suggestions made by the reviewers. One of the AR simulations was conducted by Yun-Ting Tasi, an assistant, in a city.

**Conflicts of Interest:** The authors declare no conflict of interest.

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
