# Peer review of "Physical and Augmented Dynamics of a Cultural Event"

_applsci, doi:10.3390/app12147001_

Round 1

Reviewer 1 Report

The manuscript is well structured and presented.

Even if there are no great elements of novelty in the techniques used and in their application, we believe that it can be published.

Author Response

Dear Reviewer:

On behalf of my co-author, thank you very much for this significant reviewing effort.

Your suggestions are highly appreciated.

Best regards,

Naai-Jung Shih

Reviewer 2 Report

Manuscript Number: applsci-1771407

Full Title:   Physical and Augmented Dynamics of a Cultural Event

 The work submitted to AS by Naai-Jung Shih and Tzu-Yu Chen concerns to combine the Taiwan Lantern Festival (TLF) related physical dynamics of land use and lantern installations, with the geomatic techique to augmented dynamics of installations at reallocated sites, in five cities in Taiwan for 2016 and 2020.

The paper is arranged according AS instruction for authors and is delightful to read from the point of view of the reviewer. The study shows a substantial effort on the part of authors to implement the project and its potential applications and I hope that after the review, the authors will be able to resubmit the work. Although the topics is of considerable interest to the scientific community, this reviewer observes that there are two criticisms with regard to figures and bibliographical references. Indeed, it would be appropriate for the reader to have both an indication of the geomatic characteristics (coordinates, datum, and so on) of the individual cities studied and references to appropriate works. Many of these works are in the reviewer's opinion not relevant and some are even in the original Chinese language.

I suggest changes as described below:

 1)      line 64, figure 1: please modify figure, with coordinates, datum, and so on of Taipei;

2)      line 269, please see previous comment n. 1;

3)      line 362, please see previous comment n. 1;

4)      line 521-524: these reference is missing in the url, please insert it for the reader;

5)      line 525-530, please modify, is in Chinese language, the reader expects to find an in-depth study in English, used by the scientific community;

6)      line 539-540, please see previous comment 5;

7)    line 593-602, in my opinion these references are not relevant (permafrost, mining operation, alpine terrain, forest, …), please change them in the text;

8)      line 618-619, the url is not accessible;

9)      line 646-647, please see previous comment 8.

I will be available to authors for the assessment of the manuscript at the subsequent submission.

Best regards

Author Response

(The authors gave the same response as above.)
